# Learning Retinal Representations from Multi-modal Imaging via Contrastive Pre-training

**Emese Sükei**[1]                                             EMESE.SUEKEI@MEDUNIWIEN.AC.AT
**Elisabeth Rumetshofer**[2]                                   RUMETSHOFER@ML.JKU.AT
**Niklas Schmidinger**[2]                                      SCHMIDINGER@ML.JKU.AT
**Ursula Schmidt-Erfurth**[1]              URSULA.SCHMIDT-ERFURTH@MEDUNIWIEN.AC.AT
**Günter Klambauer**[2]                                        KLAMBAUER@ML.JKU.AT
**Hrvoje Bogunović**[1]                           HRVOJE.BOGUNOVIC@MEDUNIWIEN.AC.AT
[1] *OPTIMA Lab, Department of Ophthalmology, Medical University of Vienna, Austria*
[2] *LIT AI Lab, Institute for Machine Learning, Johannes Kepler University, Linz, Austria*

**Editors:** Accepted for Poster Presentation at MIDL 2023

## Abstract

Contrastive representation learning techniques trained on large multi-modal datasets, such as CLIP and CLOOB, have demonstrated impressive capabilities of producing highly transferable representations for different downstream tasks. In the field of ophthalmology, large multi-modal datasets are conveniently accessible as retinal imaging scanners acquire both 2D fundus images and 3D optical coherence tomography (OCT) to evaluate the disease. Motivated by this, we propose a CLIP/CLOOB objective-based model to learn joint representations of the two retinal imaging modalities. We evaluate our model's capability to accurately retrieve the appropriate OCT based on a fundus image belonging to the same eye. Furthermore, we showcase the transferability of the obtained representations by conducting linear probing and fine-tuning on several prediction tasks from OCT.

**Keywords:** contrastive learning, predictive modelling, multi-modal imaging, retina

## 1. Introduction

Self-supervised learning aims to learn representations without manual labelling, often through contrastive or reconstructive tasks, enabling efficient downstream task learning with fewer annotated labels. In medical imaging, learning a meaningful representation by jointly modelling different imaging modalities can facilitate disease progression modelling and personalised patient management. In retinal imaging, combining 2D fundus photography or near-infrared reflective imaging with 3D optical coherence tomography (OCT) is readily available and can provide complementary information about the retina's structure. However, existing multi-modal methods in ophthalmology are fusion-based and rely on supervised learning signals (Jin et al., 2022), while unsupervised multi-modal contrastive representation learning in this field remains largely under-explored.

To address this gap, we propose a multi-modal pre-training method based on contrastive objectives (CLIP (Radford et al., 2021) or CLOOB (Fürst et al., 2022)) to learn proficient OCT and fundus image encoders without the need for expert annotations. We show that our method can provide both a retrieval system and encoders to obtain comprehensive OCT and fundus image representations for several downstream tasks.

SÜKEI[1] RUMETSHOFER[2] SCHMIDINGER[2] SCHMIDT-ERFURTH[1] KLAMBAUER[2] BOGUNOVIĆ[1]

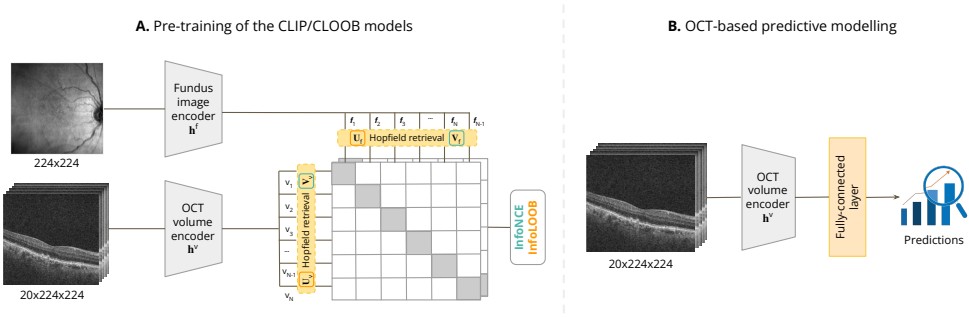

Figure 1: **A.** Contrastive pre-training of the encoders of the two retinal imaging modalities using CLIP/CLOOB. **B.** Using the pre-trained OCT volume for downstream tasks.

## 2. Method

The proposed contrastive framework (Figure 1**A**) utilises CLIP and CLOOB objectives, InfoNCE and InfoLOOB, respectively. It employs ResNet18 with pre-trained ImageNet weights as the backbone image encoder and VideoResNet18 with pre-trained Kinetics (Kay et al., 2017) weights as the backbone volume encoder for fundus images and OCT volumes. The dimension of the embedding space is set to $d = 512$, which determines the output size of both encoders. The hyper-parameters and training strategies suggested by OpenCLIP (Wortsman et al., 2022) and CLOOB are used. After contrastive pre-training, the fundus encoder is discarded, and only the volume encoder is used to extract descriptive feature representations for downstream tasks. This is achieved by adding a single fully-connected layer after the encoder (Figure 1**B**). To demonstrate the models' feature extractor capabilities, linear probing is performed by freezing the encoder weights and training only the last layer. Additionally, we fine-tuned the entire model for the downstream tasks.

## 3. Experiments & Results

**Dataset and preprocessing** For pre-training the CLIP/CLOOB models, the study uses large-scale data from OPTIMA Lab imaging datasets. We extracted 70,767 fundus photography and OCT volume pairs acquired from 2,987 patients with neovascular age-related macular degeneration (nAMD) using Spectralis, Cirrus, Nidek, or Topcon scanners. As an additional *external dataset* for the downstream tasks, we use data from the HARBOR trial, which contains OCT volumetric scans of 1098 patients undergoing treatment for nAMD, with corresponding clinical and treatment labels (Busbee et al., 2013). To allow large batch sizes, we downsize the fundus images and the OCT B-scans to 224x224. For OCT volume, we then sample 20 B-scans randomly using a Gaussian probability distribution centred on the central B-scan. Finally, the images/volumes are normalised.

**Contrastive pre-training** The pre-training dataset is divided into train-validation-test sets at a ratio of 80%-15%-5%, using 3,537 fundus image-OCT volume pairs for the holdout set to evaluate the models' retrieval ability. In this set, CLIP ranked the correct OCT volume first in 10.51% of cases, while CLOOB ranked the correct OCT volume first in

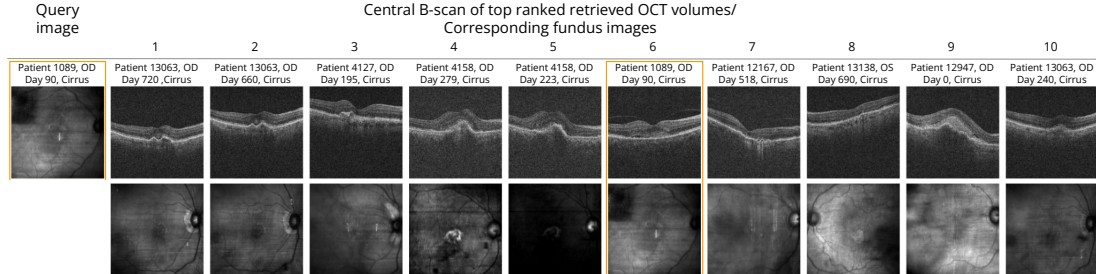

Figure 2: Example results for the retrieval task on the hold-out test set using CLOOB. Orange boxes indicate the matching fundus image - OCT volume pair.

11.36% of cases. Figure 2 provides a qualitative example of this. It's important to note that this task is close-to-impossible for human experts to perform accurately.

**External downstream tasks** We define three downstream tasks on the external dataset, namely: central subfield thickness (CST) prediction, best corrected visual acuity (BCVA) prediction (Snellen equivalent of $< 20/60$), and high treatment need (TN) forecasting. The first is a regression task, while the latter are binary classification tasks. To evaluate the models' performance, we use a 5-fold cross-validation technique, where we split the external dataset into train-validation-test sets at a ratio of 80%-10%-10% per patient, stratified by the target outcome. Our preliminary results (Table 1) show a notable improvement using the CLIP/CLOOB pre-trained encoders over the Kinetics baseline across the different tasks.

Table 1: Results of linear probing and fine-tuning on the downstream tasks. The mean and standard deviation of the performance measures over the 5-folds are reported.

| Outcome | Initialisation | Linear probing | | Fine-tuning | |
|---|---|---|---|---|---|
| | | **AUROC** | **AUPRC** | **AUROC** | **AUPRC** |
| BCVA | Kinetics | 0.788 (0.027) | 0.717 (0.028) | 0.854 (0.033) | 0.784 (0.058) |
| | CLIP | **0.847 (0.044)** | **0.775 (0.088)** | 0.866 (0.052) | **0.812 (0.066)** |
| | CLOOB | 0.818 (0.048) | 0.739 (0.103) | **0.872 (0.029)** | 0.801 (0.053) |
| High TN | Kinetics | 0.481 (0.232) | 0.354 (0.194) | 0.811 (0.011) | 0.675 (0.021) |
| | CLIP | **0.808 (0.047)** | **0.690 (0.075)** | 0.840 (0.020) | 0.707 (0.051) |
| | CLOOB | 0.788 (0.089) | 0.606 (0.137) | **0.868 (0.031)** | **0.763 (0.069)** |
| | | **RMSE [μm]** | **R-squared** | **RMSE [μm]** | **R-squared** |
| CST | Kinetics | 114.720 (37.496) | 0.107 (0.098) | 85.368 (12.068) | 0.354 (0.185) |
| | CLIP | **97.222 (25.108)** | **0.243 (0.098)** | **71.730 (12.764)** | **0.551 (0.205)** |
| | CLOOB | 102.344 (30.081) | 0.175 (0.043) | 76.683 (17.073) | 0.535 (0.131) |

**Conclusions** Our initial findings suggest that using contrastive pre-training with multimodal retinal images yields transferable and meaningful OCT volume representations, which can be leveraged for other clinical tasks. We plan to conduct additional analysis on diverse datasets and downstream tasks to evaluate the approach's potential and limitations better.

## Acknowledgments

This work received financial support from the FWF Austrian Science Fund (grant number FG 9-N).

SÜKEI[1] RUMETSHOFER[2] SCHMIDINGER[2] SCHMIDT-ERFURTH[1] KLAMBAUER[2] BOGUNOVIĆ[1]

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
