# OpenReview forum: "Learning Retinal Representations from Multi-modal Imaging via Contrastive Pre-training"
_MIDL.io/2023/Short_Paper_Track — MIDL 2023 Short paper track Poster_

### Official Review · Reviewer_s85n · 2023-04-10
**well written and insightful**

**Rating:** 9
**Confidence:** 4

**Review:**

The authors proposed to use contrastive pre-training across imaging modalities in retinal imaging  (2D fundus, 3D OCT)  to solve difficult classification and regression tasks for the 3D OCT. For this purpose, they use modern contrastive objective (CLIP, CLOOB) built on top of modern pretrained networks. Once trained, only the encoder of the 3D OCT is kept, and a simple layer suffices to produce impressive results, compared with building on the pretrained network directly. The paper are very well written and the results are comprehensive and compelling.

My only comment is that the Acknowledgement should be in the 3rd page not to violate the page limit.

---

### Official Review · Reviewer_ine6 · 2023-04-19
**Solid results but explanation needs work**

**Rating:** 7
**Confidence:** 4

**Review:**

This paper proposes use of contrastive learning to improve prediction of clinical measures from OCT volumes. Overall this seems like a solid paper. However, it isn’t the easiest to read. None of the acronyms are spelt out, and these methods aren’t explained. While appreciating space is limited, I think the benefits of the method would be much more clearly understood if readers could gain a high level understanding for what is implemented. Similarly, what is a Hopfield network? I don’t follow this statement: ‘CLOOB with modern Hopfield networks is particularly promising for medical images due to its ability to counteract the explaining-away effect’, what explaining-away effect?